# Experimental Study on Potential Influence of the Invasive *Hedychium coronarium* J. König on the Evapotranspiration of Riparian Plant Community

**DOI:** 10.3390/plants12091746

**Published:** 2023-04-24

**Authors:** Driélli de Carvalho Vergne, Lívia Malacarne Pinheiro Rosalem, Edson Cezar Wendland, Jamil Alexandre Ayach Anache, Márcia Cristina Martins da Silva, Raquel Stucchi Boschi, Dalva Maria da Silva Matos

**Affiliations:** 1Graduate Program in Ecology and Natural Resources, Federal University of São Carlos, São Carlos 13565-905, SP, Brazil; 2Department of Hydraulics and Sanitary Engineering, São Carlos School of Engineering, University of São Paulo, São Carlos 13560-250, SP, Brazil; 3Faculty of Engineering, Architecture and Urbanism and Geography, Federal University of Mato Grosso do Sul, Campo Grande 79070-900, MS, Brazil; 4Environmental Management and Sustainability, Federal University of São Carlos, São Carlos 13565-905, SP, Brazil; 5Department of Hydrobiology, Federal University of São Carlos, São Carlos 13565-905, SP, Brazil

**Keywords:** dominance, exotic species, herbaceous invader, tropical forest

## Abstract

The balance between precipitation and evapotranspiration (ET) has direct effect on vegetation, and any change in its structure and composition can influence it. The aim of this study is to determine experimentally the daily evapotranspiration (ET) of the invasive species, *Hedychium coronarium,* and to compare with a group of four native species of the riparian forest. The experiment was carried out in a greenhouse with three different treatments: (1) only the invasive species; (2) only native species; and (3) a mixture of invasive and native species. In each lysimeter, pressure transducers recorded the water level at every 15 min along 14 months. Daily ET was calculated by the method of Gribovszki et al. (2008) and varied according to the treatment, indicating that different species (invasive or native) use the water differently. The maximum accumulated daily ET occurred for mixture treatment (2540.16 mm), while the treatment with the invasive plant presented the lowest value (2172.53 mm). *H. coronarium*, in monodominant stands, can reduce evapotranspiration on invaded areas and increase it when immersed in the riparian forest.

## 1. Introduction

Invasive plant species are the main drivers of change at ecosystemic level [1], such as hydrological cycle [2], in addition to being one of the main causes of biological diversity loss [3]. The success of invasive species relies on their biological attributes which allow them to grow faster than native species [4] but also, on the environmental conditions which would be favorable to their growth [5].

Invasive plant species generally use a large amount of water in their life cycle, mainly because of their faster and continuous growth [6] and higher evaporation rate when compared to native species [7,8]. Vegetation structure directly affects the balance between precipitation and evapotranspiration, as it depends directly on the intensity of the radiation, the availability of water in the soil and the speed of the wind [9]. Thus, the presence of these species may cause negative impacts on water regime, either on the quality or the quantity of water available and direct impacts on local biodiversity and even on human life.

Currently, one of the ecosystems that has undergone major replacement of native by invasive vegetation and consequently has suffered extreme transformations are the riparian forests [10,11,12]. Riparian forests are considered a fundamental habitat for the maintenance of numerous ecosystem services, including the supply of useful products and biodiversity, as they are a source of food for aquatic and terrestrial fauna and act as a corridor for fauna [13]. Also, these forests are responsible for filtering pollutants, pesticides and sediments that reach water bodies [14,15]. However, because most resources are highly available, especially light and water, invasive species are spreading fast.

In Brazilian riparian forests, where trees are the dominant group over herbaceous and shrubs, the spread of the invasive macrophyte *Hedychium coronarium* is causing impacts at the community level by affecting the native fauna and flora [16]. *H. coronarium* forms monodominant stands on the transition zone, from aquatic to terrestrial habitats, outcompeting native species of different growth form. It also occurs, but it is less abundant, just few amounts of small ramets, under the riparian forest. However, further disturbances in these areas could increase the spatial distribution of *H. coronarium* and consequently changing the community structure towards a monodominant vegetation. Changes on vegetation structure affect evapotranspiration [17] as it reflects the interaction among the vegetation, soil, and atmosphere [18]. Thus, the aim of this study was to understand how the invasive *H. coronarium* may affect the evapotranspiration in riparian forests through an experimental design simulating three vegetation structures: non invaded riparian forest, invaded riparian forest with the presence of *H. coronarium* and a monodominant stand of *H. coronarium*.

## 2. Results

There is a pattern in daily ET and temperature over the studied period for the three treatments (Figure 1). In addition, there was a moderate correlation between the variable medium temperature and ET (ρ = 0.49; *p* <0.001).

There was a difference in the daily ET among the treatments (Invasive and Native; Invasive and Mixture; Native and Mixture) (Figure 1 and Table 1). For the experiment containing only the invasive species the daily ET values were between 0.59 mm d^−1^ and 17.16 mm d^−1^ for average temperatures of 20.6 °C and 30.5 °C, respectively; for the treatment with only native species, values were between 1.07 and 18.27, and 16.5 °C and 30.5 °C, respectively; and for the mixture experiment the daily ET were between 0.99 and 18.50, and 19.4 °C and 30.3 °C, respectively. Thus, the minimum, average and maximum ET values of the three treatments studied were remarkably close (Figure 1) while the maximum ET values, for the three treatments, were concentrated in the month of February, in the summer. The mixture (invasive + natives) has an accumulated ET final value higher than the other two treatments (Figure 1a).

## 3. Discussion

Evapotranspiration is the way of returning water to the atmosphere, consequently, a fundamental part of the hydrological cycle. In this study we compared the evapotranspiration among three vegetation type: (1) the invasive herbaceous species (*H. coronarium*), (2) native species (Bromeliaceae, *Euterpe edulis*, *Tapirira guianensis* and *Croton urucurana*) from riparian forest and (3) a mixture with all species. The invasive vegetation had lower evapotranspiration than the natives and, than the mixture of invasive and natives. This result indicates the impact caused by *H. coronarium* on the water provision, an ecosystem service of significant importance.

In fact, more heterogeneous vegetation, especially with taller tree species (here, our simulated native and mixed treatments), tend to have a higher rate of evapotranspiration than environments with more homogeneous and lower vegetation, such as grasses, shrubs, and monocultures [19]. In general, larger tree species have larger leaves, larger leaf area, deeper roots and they are more efficient at absorbing water [20], and consequently have higher rates of ET. In addition, darker vegetation, represented here by native species, reflect less solar radiation than lighter vegetation, such as *H. coronarium*; therefore, native species have a higher ET [21]. Our experiment containing only the native species obtained higher values of daily ET, when compared to the treatment containing only the invasive species. A study showed a potential ET of 4.4 mm/d for a riparian forest where the tree strata was responsible for the highest percentage (88%) and the herbaceous for the lowest (3%) of the total value [22]. Thus, herbaceous species tend to use a small amount of water when compared to tree species. In our study, the mixed vegetation presented the highest values of ET among the three treatments, as all species grew in the same lysimeter.

Daily evapotranspiration was related to the daily temperature and varied among treatments, indicating that each vegetation type uses the water available in the environment in a different way. In general, ET tends to increase according to the increase in the temperature along the year, especially during the summer, from October 2018 to March 2019, when the solar radiation is also higher. This relationship was also found in a riparian forest located in Brotas municipality, about 70 km from our experiment [23]. Similarly, ref. [24] found the highest rates of ET in the summer, December and February, in sugarcane and eucalyptus plantations and native areas of Cerrado and Atlantic Forest. Furthermore, these higher ET values related to the tree strata may be due to greater height, deeper roots [25], in addition to greater biomass production. Species with higher biomass and, consequently, larger size can consume a greater amount of water [26]. These morphological characteristics are not applicable to the invasive species targeted in this study, since it is an herbaceous plant.

In the treatment containing the mixture of species (invasive and native) the minimum ET was lower when compared to the treatment containing only native species, this may have occurred due to the presence of the species *H. coronarium*. As previously discussed, this species alone had lower minimum and maximum ET, which shows a direct consequence of plant invasion in the water system. Ref. [27] found higher values of daily ET associated with places with less anthropic influence, such as native forests and reforestation. In addition, ET directly influences the climate through the exchange of energy and water, making native forests essential for ecological services and watersheds, such as flood control, water quality and carbon and nitrogen stocks [28]. In contrast, the replacement of native forest by plantations or pastures reduces the infiltration capacity of the soil, the depth of the roots and the rigidity of the surfaces, in addition to increasing the albedo [29,30]. 

Considering the change on the vegetation structure in monodominant stands by replacement of native vegetation, mostly tree species, in riparian areas by the herbaceous *H. coronarium* it would be expected to have negative impacts on the ecosystem functions and services. The description of the ET of an invasive species, as the comparison of these values with ET of native species, even at an experimental level, is necessary to better understand these processes in natural habitats. Generally, studies seek this measurement in natural environments, which is made more difficult by having many variables to be controlled. 

Although studies have shown the impact of *H. coronarium* on native species [31,32] and on riparian ecosystems in Brazil [16,33,34], this species may have a greater impact on their environment than what can be measured through changes on diversity and ecological interactions. The findings of this work show the lower evapotranspiration of an invasive species compared to native ones, which may affect the hydrological cycle. The decreased evapotranspiration in monodominant stands reflects a decreased amount of water in the air which consequently decreases the rainfall, affects the energy balance, and ultimately changes the climate along time and space. 

The increase in evapotranspiration in areas invaded by *H. coronarium* could, perhaps, be one of the consequences of the process of greater water retention in the leaves. This can be said, due to the fact that this invader, a specie rhizomatous and great producer of biomass [33,35], covers the soil in a way that less interception of water occurs. With less water percolating through the soil and more water retained in the leaves, evapotranspiration would tend to increase. Considering its morphology, *H. coronarium* has a large leaf area (long and wide leaves), which consequently increases its transpiration, due to the increase in the surface area for gas exchange. Also in this sense, this plant has a high photosynthetic rate, which can also lead to an increase in the volume of water that is absorbed and evapotranspirated.

The invasive *H. coronarium* is spreading in the riparian areas of the Atlantic Rainforest where air humidity is high compared to disturbed areas. Besides the impacts on diversity, we have observed changes on the water channel with high sedimentation and lateral flooding in invaded areas (personal observation). These impacts are related to the season and region (distance to the water channel). How would the impact be in a longer time span? Probably worse, as the allelochemical impact observed on the native species [31] would favor the spread of this species towards the forest, decreasing the establishment of native trees. 

As our results are very similar to those observed for other combinations of herbaceous and tree species in natural habitats, we may use them to simulate the decrease of evapotranspiration in natural invaded areas and consequently contribute to estimate changes on rainfall and water table recharge. Thus, we may have positive feedback between this invasive species and climate change, as it has caused precipitation and evapotranspiration changes, which would favor its spread while decreasing the regeneration of tropical forests.

## 4. Materials and Methods

### 4.1. Methodology

*Hedychium coronarium* J. König (Zingiberaceae) is a perennial macrophyte that occurs in both humid environments and understory in riparian forest [36], that has been commonly found in Brazilian riparian forests, where it is spreading fast by changing soil properties, nutrient cycling, and outcompeting native species [16,33,34]. To compare the evapotranspiration (ET) of the invasive species *H. coronarium* and native species, a controlled experiment was carried out inside a greenhouse, under natural and temperature conditions, located in the campus of the Federal University of São Carlos (UFSCar), municipality of São Carlos, Brazil (22° 0′ 55″ S, 47° 53′ 28″ W). 

The experiment was carried out by assembling lysimeters (Figure 2), which were tanks used to accurately measure evaporation, precipitation, and drainage events [37]. Six lysimeters in total were assembled using 1000 L plastic tanks (smaller diameter: 1.14 m; larger diameter: 1.44 m and height: 0.78 m). In each lysimeter we added gravel (5 cm), coarse sand (5 cm) and a mixture of vegetal soil (60%) and horizon A2 (40%), until the box was fulfilled. After that, we prepared two replicates of each of the following treatments, simulating three vegetation types: (1) a totally invaded area containing only the invasive plant, where eight rhizomes of *H. coronarium* were planted, hereafter called Invasive; (2) an uninvaded area containing eight individuals of native species of different growth form: two individuals of a herbaceous (Bromeliaceae) and two of each tree species commonly found in riparian forests: *Euterpe edulis* Mart., *Tapirira guianensis* Aubl. and *Croton urucurana* Baill., hereafter called Native; (3) an invaded area containing the invasive and the same native species used in treatment 2, where we planted four individuals of *H. coronarium* and one individual of each of the four native species, hereafter called Mixture. Each tree individual had an average height of 60 cm in the planting moment to guarantee their establishment.

In order to manually measure the water level in each treatment, we installed a PVC tube (32 mm) within each treatment, laterally to the water tank, to which we attached a 1/8 inch crystal hose and a graduated measuring tape. Also, in the center of each treatment we installed another PVC tube (32 mm) previously covered with a geotextile blanket and blocked at one end with a “cap”. Each central tube was placed for continuous monitoring of the top-down water level by the pressure transducer present in each treatment. A TD-diver (pressure transducer) (DI-801, Van Essen Instruments) was placed in each treatment and a Baro-diver (barometric pressure transducer) (DI-800, Van Essen Instruments) was allocated inside the greenhouse, so that the local atmospheric pressure could be measured. Both TD-diver and Baro-diver have accuracy maximal of ±2.0 cm H_2_O and accuracy typical of ±0.5 cm H_2_O. 

The TD-diver is a submersible “datalogger” that takes measurements at predefined time intervals in the long term, being able to monitor the water level in real time using a pressure sensor. The pressure sensor measures the equivalent hydrostatic pressure of the water above the sensor diaphragm, which calculates the total water depth. The TD-diver independently measures pressure (cm of the water column), temperature (°C) and records them in its internal memory. The Baro-diver is also a “datalogger” for real-time monitoring; however, it records the atmospheric pressure (cm of the water column) in addition to the temperature (°C). 

The water level in each lysimeter was calculated by discounting the atmospheric pressure value from the hydrostatic pressure value. Each TD-diver was set up to record level measurements every 15 min. The configuration of the central tubes guaranteed the passage of water along the transducers only, and the location of the tubes guaranteed lower temperature fluctuations of the water. This was necessary to avoid any interference of the variation in the greenhouse temperature and on the measurements made by the divers. 

All treatments were manually watered, and the amount of water used in each lysimeter was quantified. The water level of the lysimeters was monitored daily, around 9:30 am, using the measuring tape attached to the transparent hose. When the level was less than or equal to 20 cm, we added water until this level reached approximately 45–47 cm in the hose (maximum height of the soil in each lysimeter). Such manual measures were performed as a control measure, to compare manual and divers’ measurements along 14 months, totaling 398 days. Meteorological data, such as solar radiation and relative humidity were obtained from the National Institute of Meteorology (INMET), at São Carlos station (−21.98, −47.88), which is located at approximately 1.1 km from the location of the experiment. The temperature was registered by the pressure transducer (Baro-diver) while we adopted a constant wind speed of 0.5 m s^−1^ within the greenhouse.

### 4.2. Calculation of Evapotranspiration (ET)

To calculate ET we adapted the method of Gribovski et al. (2008) (Equation (1)):P − ET_d_ − Q = ΔS,(1)
where P is the precipitation (mm d^−1^); ET_d_ is the daily evapotranspiration (mm d^−1^); Q is the side recharge; and ΔS (mm d^−1^) is the variation of water in a controlled volume. As our experiment was carried out in a greenhouse, the amount of water placed in each lysimeter was controlled as there was no precipitation inside. In addition, there was also no side recharge due to the lysimeters being built in 1000 L boxes. Thus, the adjusted equation (Equation (2)):ET_d_ = ΔS,(2)
where the daily evapotranspiration (ET_d_) depends only on the variation of stored volume.

As the variation of the water level in a controlled volume depends, among other factors, on the soil’s porosity, it was necessary to obtain the specific yield values for each lysimeter. Specific yield is the percentage of water that is free to drain along the soil under the influence of gravity. The calculation of specific yield (Sy) was performed using Equation (3):Sy = V_d_/V_t_×100,(3)
where V_d_ is the total volume drained and V_t_ is the total volume of the lysimeter. The specific yield value calculated for each lysimeter is presented at Table 2.

Since the ET calculation depends on the specific yield (Sy), the final Equation (4) is:ET = Sy × ΔS,(4)
where ET is evapotranspiration; Sy is the specific yield; and ΔS is the variation of water in a controlled volume.

Thus, we obtained the ET of each lysimeter submitted to the different treatments (two lysimeters in each of the three treatments, totaling six) with the variation of the controlled volume (ΔS) obtained by the pressure transducers and the final value of the specific yield (Sy). Due to the conical shape of the lysimeters, the calculation of the variation of the controlled volume (ΔS) was performed by calculating the volume of water corresponding to the variation of the level indicated by the divers. With the results every 15 min of the water level by the pressure transducers, the ET calculation was made, being later grouped into daily and monthly values.

### 4.3. Statistical Analysis 

Tukey test and the Mixed Models (Linear Mixed Effects Regression—LMER) with the ‘lmer4′ package [38], were applied to assess possible differences in the daily ET (response variable) between the three treatments tested (Invasive, Natives and Mixture) (explanatory variable). We used mixed models as it allows greater freedom in the combination of variables. Therefore, the relationship between the response and explanatory variables was the fixed effects of the model, while replicates of treatments and collection dates were considered as the random effect. To make sure that the manual measurements (tape measure) and pressure transducers were equivalent, we performed a regression with the data below 30 cm from both measurements. All statistical analyzes were performed in the R environment [39].

## 5. Conclusions

In this study, we measured the evapotranspiration of three groups of species: one invasive herbaceous species (*Hedychium coronarium*), a combination of four native species (Bromeliaceae, *Euterpe edulis, Tapirira guianensis* and *Croton urucurana*), and a mixture of invasive and native species. All species occur in riparian forests and compete directly in the same environment. Measurements were made using a self-made lysimeter made of tanks that accurately measured evaporation, precipitation, and drainage events. The invasive herbaceous species (*H. coronarium*) had lower evapotranspiration compared to both the group of native species (Bromeliaceae, *Euterpe edulis, Tapirira guianensis* and *Croton urucurana*) and the mixture of invasive and native species. On average, *H. coronarium* presented a daily ET rate 45% lower than that of native species and 41% lower than that of the mixed group. As per the results, evapotranspiration was significantly lower in natural areas invaded by *Hedychium coronarium*. The values obtained through this study may be used to simulate changes in rainfall and water table recharge.

## Figures and Tables

**Figure 1 plants-12-01746-f001:**
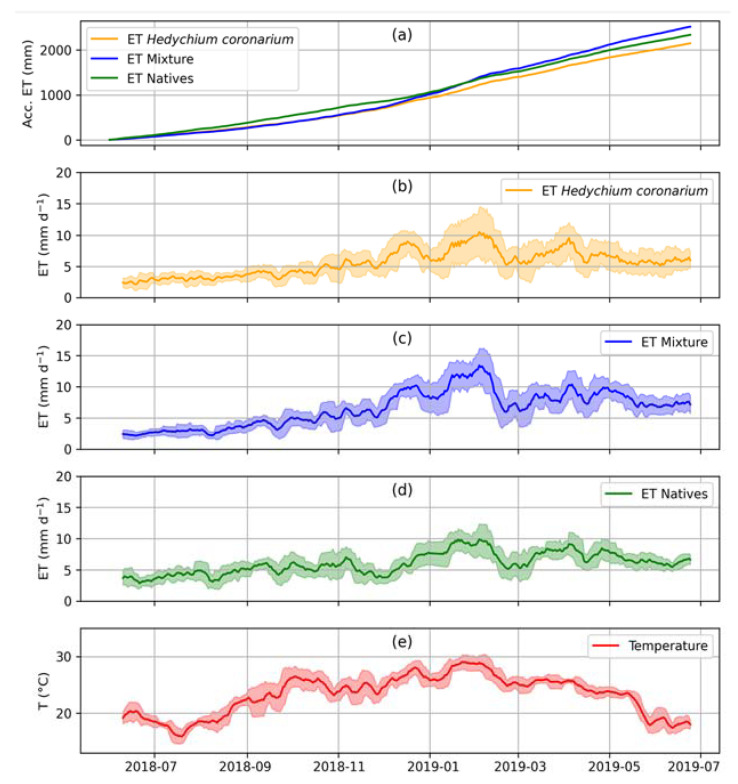
(**a**) Graphical representation of the accumulated daily evapotranspiration pattern (mm d^-1^) over 14 months of study, for the three treatments (invasive, mixture, and natives); (**b**) ET from lysimeters containing only the invasive *Hedychium coronarium*; (**c**) ET from a mixture of the invader and the natives; (**d**) ET from lysimeters containing only native species of riparian forest, Bromeliaceae, *Euterpe edulis*, *Tapirira guianensis* and *Croton urucurana*; and (**e**) average temperature (°C) over 14 months of study.

**Figure 2 plants-12-01746-f002:**
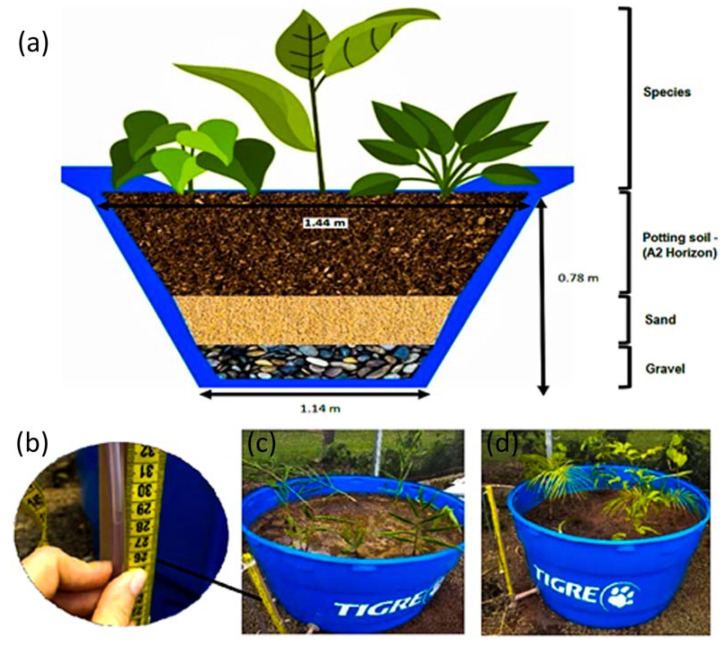
(**a**) Scheme of the lysimeters used in the evapotranspiration experiment; (**b**) checking the water level on a graduated tape in the pvc pipe attached to the lysimeter; (**c**) tank with the invasive species (*H. coronarium*); (**d**) tank with the native species. (**b**–**d**): photos taken by authors in a greenhouse at the Secretary of Environmental Management and Sustainability of the Federal University of São Carlos, Brazil.

**Table 1 plants-12-01746-t001:** Tukey test comparing daily evapotranspiration (mm/d) three different treatments: (1) lysimeters containing only the invasive *Hedychium coronarium*; (2) lysimeters containing only the native species of the riparian forests- Bromeliaceae, *Euterpe edulis*, *Tapirira guianensis* and *Croton urucurana* (Native); and (3) a mixture of invasive and native species (Mixture). All lysimeters were kept in a greenhouse and monitored monthly over 14 months.

	Estimate	SE	df	t	*p*
*Hedychium coronarium* × Mixture	−0.924	0.107	1590	−8.616	<0.001 *
*Hedychium coronarium* × Native	−0.498	0.107	1590	−4.649	<0.001 *
Native × Mixture	0.425	0.107	1590	3.967	<0.001 *

* Significative values.

**Table 2 plants-12-01746-t002:** Values of specific yield (Sy), in percentage, V_t_ (the total volume of each lysimeter) and V_d_ (volume drained), both in liters (L). Invasive I and II are the treatments with *Hedychium coronarium* only, Native I and II had only native species and Mixture I and II had both invasive and native species.

Treatment	Vt (L)	Vd (L)	Sy (%)
Invasive I	554.79	32	5.86
Invasive II	569.72	45	7.90
Native I	533.61	55	10.31
Native II	574.11	55	9.58
Mixture I	624.38	60	9.61
Mixture II	587.83	50	8.51

## Data Availability

Data used in this study can be available by contacting at dmatos@ufscar.br.

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
