# Peer review of "Experimental Study on Potential Influence of the Invasive Hedychium coronarium J. König on the Evapotranspiration of Riparian Plant Community"

_plants, 2023, doi:10.3390/plants12091746_

Round 1

Reviewer 1 Report

The manuscript describes the evapotranspiration process of a riparian system in the presence of the invasive species Hedychium coronarium. The results indicate that based on the experimental design, a negative effect of the presence of H. coronarium is not appreciated.

The evapotranspiration process involves the components of plant transpiration, water evaporation and possible replacement by precipitation, however, in this paper no mention is made of the transpiration records in the tissues of the evaluated species. It would be of great importance to have considered this record to make ecophysiological inferences about the interaction of the evaluated species. I suggest that in case of having the transpiration data they be included to complement the results and the discussion.

On the other hand, in the methodology it is not very clear how the device was designed to evaluate evapotranspiration, since it is mentioned that it was an experiment in a greenhouse using juvenile individuals of the trees. In this case, a figure or a representative photograph could be very helpful.

Based on the description of the system used to evaluate evapotranspiration, it should be considered in the discussion that the initial growth of the tree seedlings and the rhizomes of H. coronarium was probably not significant due for the large volume of the tanks used.

Additionally, it is not clear how many replicates the experiment had to be able to apply the Tukey test shown in Table 1.

Finally, I suggest that a section of conclusions of the work be included where the conclusions based on the results obtained in this work are clearly and concisely indicated.

Author Response

Thank you for your suggestions. They all were included in the revised version of the ms.

  • The manuscript describes the evapotranspiration process of a riparian system in the presence of the invasive species Hedychium coronarium. The results indicate that based on the experimental design, a negative effect of the presence of coronarium is not appreciated.

The evapotranspiration process involves the components of plant transpiration, water evaporation and possible replacement by precipitation, however, in this paper no mention is made of the transpiration records in the tissues of the evaluated species. It would be of great importance to have considered this record to make ecophysiological inferences about the interaction of the evaluated species. I suggest that in case of having the transpiration data they be included to complement the results and the discussion.

Unfortunately, we don’t have data about plant transpiration.

  • On the other hand, in the methodology it is not very clear how the device was designed to evaluate evapotranspiration, since it is mentioned that it was an experiment in a greenhouse using juvenile individuals of the trees. In this case, a figure or a representative photograph could be very helpful.

We included a figure (Figure 2) showing the experiment design. We believe that with it this point becomes clearer. We take the opportunity and comment to also improve the quality of Figure 1 (in the Results), without changes in values, just making the graphics better visualized.

  • Based on the description of the system used to evaluate evapotranspiration, it should be considered in the discussion that the initial growth of the tree seedlings and the rhizomes of coronarium was probably not significant due for the large volume of the tanks used.

Additionally, it is not clear how many replicates the experiment had to be able to apply

the Tukey test shown in Table 1.

Each treatment was performed in duplicate; Tukey's test was applied comparing the six tanks daily (total of 398 days).

  • Finally, I suggest that a section of conclusions of the work be included where the conclusions based on the results obtained in this work are clearly and concisely indicated.

We thank you for this suggestion and have added the conclusions section at the end of the manuscript.

Reviewer 2 Report

The authors present an interesting experimental study about the effects of an invasive species that occupies a riparian plant community in tropical forests. Centered on the ET, through an experimental essay, the authors confirm that those plots with invasive species (the herbaceous H. coronarium) rather than mixture or native, have lower ET than the other plots indicating a impact on ecosystems services such water provision among others.

The experiment is well designed and the methodology appropiate for obtaining the desired results.

This class of projects could arise interest of the audience who develop experimental studies on invasive species indoor.

The manuscript is well written and properly organized. Anyway, to become a paper for Plants I suggest a couple of minor changes.

In table 1 (page 3), the p value of Native-Mixture is 0.0002, I suggest that maybe it could be <0.001 as the rest.

In table 2 (page 6) maybe you could order by treatments (i.e. Invasive I, Invasive II,....).

So, it's all. Congratulations for the research, I've enjoyed reading it.

Author Response

Thank you for your suggestions. They all were included in the revised version of the ms.

  • The authors present an interesting experimental study about the effects of an invasive species that occupies a riparian plant community in tropical forests. Centered on the ET, through an experimental essay, the authors confirm that those plots with invasive species (the herbaceous coronarium) rather than mixture or native, have lower ET than the other plots indicating an impact on ecosystems services such water provision among others.

The experiment is well designed and the methodology appropriate for obtaining the desired results. This class of projects could arise interest of the audience who develop experimental studies on invasive species indoor.

The manuscript is well written and properly organized. Anyway, to become a paper for Plants I suggest a couple of minor changes.

We all thank you for your consideration about the contribution of our paper.

  • In table 1 (page 3), the p value of Native-Mixture is 0.0002, I suggest that maybe it could be <0.001 as the rest.

We standardized the p values in this revised version.

  • In table 2 (page 6) maybe you could order by treatments (i.e. Invasive I, Invasive II,....).

Accepted and done.

  • So, it's all. Congratulations for the research, I've enjoyed reading it.

Thank you again!

Round 2

Reviewer 1 Report

The authors made the recommendations of the review.